

# Incidence of acute kidney disease after receiving hematopoietic stem cell transplantation: a single-center retrospective study

Akira Mima[1], Kousuke Tansho[1], Dai Nagahara[1] and Kazuo Tsubaki[2]

[1] Department of Nephrology, Kindai University Faculty of Medicine, Kindai University Nara Hospital, Nara, Japan
[2] Department of Hematology, Kindai University Faculty of Medicine, Kindai University Nara Hospital, Nara, Japan

## ABSTRACT

**Background:** Previous reports have shown that acute kidney injury (AKI) is common after hematopoietic stem cell transplantation (HSCT), which is a crucial treatment for patients with hematological disorders. AKI could increase mortality and induce adverse effects including the development of chronic kidney disease. The incidence of AKI in association with HSCT reportedly varies significantly because several definitions of AKI have been adopted. Acute kidney disease (AKD) is a new concept that can clinically define both AKI and persistent decreases in glomerular filtration rate (GFR) state. We conducted a retrospective cohort study to determine the incidence of AKD after HSCT.
**Methods:** This study included 108 patients aged between 16 and 70 years undergoing HSCT. In this study, AKD included clinical condition of AKI or subacute decreases in GFR. AKI was defined according to the Kidney Disease: Improving Global Outcomes guidelines based on serum creatinine. However, urine output data were not included to define AKI because the database lacked some of these data. Comparisons were made between groups using the Mann–Whitney U test.
**Results:** Acute kidney disease occurred in 17 patients (15.7%). There were significant differences between the AKD and non-AKD with respect to ABO-incompatible HSCT ($p = 0.001$) and incidence of acute graft versus host disease (GVHD) after HSCT ($p < 0.001$). The 100-day overall survival of patients with AKD and without AKD after HSCT was 70.6% and 79.8%, respectively ($p = 0.409$).
**Discussion:** ABO-incompatible HSCT and acute GVHD after HSCT were risk factors for the incidence of AKD. However, we could not find a significant association between AKD after HSCT and mortality.

Corresponding author
Akira Mima,
amima@med.kindai.ac.jp

## INTRODUCTION

Hematopoietic stem cell transplantation (HSCT) is a fundamental therapy for high risk and refractory hematological diseases. Several types of HSCT procedures exist based on

conditioning therapy (myeloablative or reduced intensity), donor source (autologous and allogenic, including matched related, matched unrelated, haploidentical and umbilical cord), and stem cell source (peripheral blood stem cells or bone marrow stem cells) (*Klumpp, 1991*; *Sawinski, 2014*). Acute kidney disease (AKD) can lead to chronic kidney disease (CKD) with increased all-cause mortality. Although acute kidney injury (AKI) is a well-defined concept, AKD definition is relatively new (*Kidney Disease Improving Global Outcomes (KDIGO), 2012*). To diagnose AKI more accurately, the new concept of AKI definition has been proposed by the Kidney Disease Improving Global Outcomes (KDIGO), which is based on and modified by the risk, injury, failure, loss of function, end-stage renal disease (RIFLE) classification and the acute kidney injury network (AKIN) criteria (*Kidney Disease Improving Global Outcomes (KDIGO), 2012*; *Akcan-Arikan et al., 2007*; *Ricci, Cruz & Ronco, 2008*; *Mehta et al., 2007*). AKD is defined as subacute reduction in glomerular filtration rate (GFR), including AKI clinical states (*Chu et al., 2014*; *Chawla et al., 2017*).

Previous reports have shown that AKI is relatively common after HSCT, and it could increase mortality and induce adverse effects including the development of CKD (*Lopes, Jorge & Neves, 2016*). Previous reports indicated that the incidence of AKI in association with HSCT ranged from 20% to 73% (*Kogon & Hingorani, 2010*), while other studies indicated 42–92% (*Clajus et al., 2012*). Study results based on these data have been varied because several definitions of AKI have been adopted.

Acute graft versus host disease (GVHD) occurs within the first 3 months after HSCT and involves the reactivity of donor immune cells against host tissues (*Filipovich et al., 2005*). Therefore, when evaluating kidney damage in acute GVHD, the conventional definition of AKI, in which AKI should be diagnosed within 1 week, may not be suitable. Thus, we used the definition of AKD, which is a new concept and could clinically define both AKI and persistent decreases in GFR state. We conducted a retrospective cohort study to determine the incidence of AKD after HSCT.

## MATERIALS AND METHODS

### Patient groups

All procedures performed in studies involving human participants were in accordance with the ethical standards of the institutional and/or national research committee and with the 1964 Helsinki declaration and its later amendments or comparable ethical standards. This is a retrospective medical record review study. The retrospective waiver of consent was obtained from the clinical study ethics review board of Kindai University Nara Hospital (approval number: 18-6).

This study included a retrospective cohort of 108 patients who underwent HSCT at Kindai University Nara Hospital from May 2006 to April 2016. Data were collected and analyzed retrospectively using electronic medical records maintained by the Department of Nephrology and hematology at Kindai University Nara Hospital. Data including year of HSCT, HSCT characteristics (e.g., underlying disease, donor type, total body irradiation, ABO incompatibility, and calcineurin inhibitor-based GVHD prophylaxis), indications for HSCT, acute GVHD, AKD, and survival outcomes were obtained from the electronic medical records and transplant database.

## Definition of AKD

Acute kidney disease was defined to include clinical condition of AKI or subacute decreases in GFR (i.e., GFR < 60 ml/min/1.73 m$^2$ for less than 3 months or decrease in GFR by ≥35% or increase in serum creatinine by >50% for less than 3 months) (*Chu et al., 2014*; *Chawla et al., 2017*). AKI was defined according to KDIGO guidelines based on serum creatinine (*Kidney Disease Improving Global Outcomes (KDIGO), 2012*). However, urine output data were not included in our definition of AKI because some data were not available in the database.

## Statistical analysis

Continuous variables were presented as medians with interquartile ranges (IQRs). Comparisons were made between groups using the Mann–Whitney U test; categorical variables were presented as numbers (percentage) and compared using the Fisher's exact tests. According to clinical features and previous reports which indicated the association with AKD after HSCT, 12 variables (age, sex, height, weight, underlying disease, with or without hypertension, with or without diabetes mellitus, total body irradiation, ABO compatibility, acute GVHD, calcineurin inhibitor-based GVHD prophylaxis, and donor and cell source) were eligible for this study.

The 100-day overall survival of patients with or without AKD after HSCT was estimated using the Kaplan–Meier method and compared using the log-rank test. All analysis was performed using StatView (SAS Institute, Cary, CA, USA). Statistical significance was defined as $p < 0.05$.

## RESULTS

We determined the clinical characteristics of 108 patients who underwent HSCT. The median age of the study participants was 49 (IQRs; 16–70) years, and 39 (36.1%) were women. The median height was 166 (IQRs; 149–184) cm and body weight was 58 (IQRs; 35–95) kg. Underlying diseases were as follows: acute lymphoblastic leukemia (ALL: 15 cases, 13.9%), acute myeloblastic leukemia (AML: 43 cases, 39.8%), chronic myeloblastic leukemia (CML: four cases, 3.7%), myelodysplastic syndromes (MDS: 10 cases, 9.3%), multiple myeloma (MM: five cases, 4.6%), aplastic anemia (AA: six cases, 5.6%), and others (25 cases, 23.1%). Hypertension and diabetes were observed in 12 (11.1%) and nine cases (8.3%), respectively. Body irradiation more than eight Gy was observed in 79 cases (73.1%). ABO-incompatible HSCT was performed in 18 cases (16.7%). Acute GVHD after HSCT occurred in 17 cases (15.7%) and calcineurin-based GVHD prophylaxis was given in 67 cases (62.0%). Donor and cell sources were as follows: allogenic bone marrow transplantation (Allo-BMT: 20 cases, 18.5%), unrelated umbilical cord blood transplantation (UR-CBT: 38 cases, 35.2%), unrelated bone marrow transplantation (UR-BT: 26 cases, 24.1%), allogenic peripheral blood stem cell transplantation (Allo-PBSCT: 11 cases, 10.2%), and autologous peripheral blood stem cell transplantation (Auto-PBSCT: 16 cases, 14.8%). AKD developed in 17 cases (15.7%) within 100 days of HSCT. Furthermore, four out of 17 AKD cases required hemodialysis. We assessed the association of AKD with the clinical characteristics

**Table 1 Univariable association between patient characteristics and post-hematopoietic stem cell transplantation.**

| Variables | All patients (n = 108) | Acute kidney disease | | p-value |
|---|---|---|---|---|
| | | Absent (n = 91) | Present (n = 17) | |
| Age (years) | 49 (16–70) | 49 (17–70) | 49 (16–66) | 0.574 |
| Female gender | 39 (36.1%) | 34(31.5%) | 5(4.6%) | 0.594 |
| Height (cm) | 166 (149–184) | 166 (150–184) | 168 (149–176) | 0.823 |
| Weight (kg) | 58 (35–95) | 59 (35–95) | 53.5 (43–77) | 0.232 |
| Underlying disease | | | | |
| ALL | 15 (13.9%) | 13 (12.0%) | 2 (1.9%) | 0.288 |
| AML | 43 (39.8%) | 37 (34.2%) | 6 (5.6%) | 0.592 |
| CML | 4 (3.7%) | 4 (3.7%) | 0 (0%) | 0.999 |
| MDS | 10 (9.3%) | 7 (6.5%) | 3 (2.8%) | 0.371 |
| MM | 5 (4.6%) | 5 (4.6%) | 0 (0%) | 0.999 |
| AA | 6 (5.6%) | 4 (3.7%) | 2 (1.9%) | 0.265 |
| Others | 25 (23.1%) | 24 (22.2%) | 1 (0.9%) | 0.063 |
| HTN | 12 (11.1%) | 2 (1.9%) | 10 (9.3%) | 0.999 |
| DM | 9 (8.3%) | 7 (6.5%) | 2 (1.9%) | 0.628 |
| TBI ≥ 8 Gy | 79 (73.1%) | 67 (62.0%) | 12 (11.1%) | 0.726 |
| ABO incompatible | 18 (16.7%) | 10 (9.3%) | 8 (7.4%) | 0.001 |
| Acute GVHD | 17 (15.7%) | 7 (6.5%) | 10 (9.3%) | <0.001 |
| CNI-based GVHD prophylaxis | 67 (62.0%) | 56 (51.2%) | 11 (0.2%) | 0.999 |
| Donor and cell source | | | | |
| Allo-BMT | 20 (18.5%) | 17 (15.7%) | 3 (2.8%) | 0.999 |
| UR-CBT | 38 (35.2%) | 31 (28.7%) | 7 (6.5%) | 0.589 |
| UR-BMT | 26 (24.1%) | 21 (19.4%) | 5 (4.6%) | 0.550 |
| Allo-PBSCT | 11 (10.2%) | 10 (9.3%) | 1 (0.9%) | 0.999 |
| Auto-PBSCT | 16 (14.8%) | 15 (13.9%) | 1 (0.9%) | 0.458 |

Note:
ALL, acute lymphoblastic leukemia; AML, acute myeloblastic leukemia; CML, chronic myeloblastic leukemia; MDS, myelodysplastic syndromes; MM, multiple myeloma; AA, aplastic anemia; HTN, hypertension; DM, diabetes mellitus; TBI, total body irradiation; GVHD, graft versus host disease; CNI, calcineurin inhibitor; Allo, allogenic; BMT, bone marrow transplantation; UR, unrelated; CBT, umbilical cord blood transplantation; PBSCT, peripheral blood stem cell transplantation; Auto, autologous.

as mentioned above. There were significant differences between the AKD and non-AKD with respect to ABO-incompatible HSCT ($p = 0.001$) and incidence of acute GVHD after HSCT ($p < 0.001$). However, we could not find a significant association between AKD and other clinical characteristics (Table 1). Then, we analyzed the 100-day overall survival of patients with AKD and without AKD after HSCT. The overall survival rate was 70.6% and 79.8%, respectively ($p = 0.409$). Thus, we could not find a significant association between HSCT-induced AKD and mortality (Fig. 1).

## DISCUSSION

This retrospective, single-center study focused on the incidence of AKD after HSCT. Our results could offer the probability of searching the risk and prognostic factor of AKD after HSCT and provide a deeper understanding of this disorder. We found that the

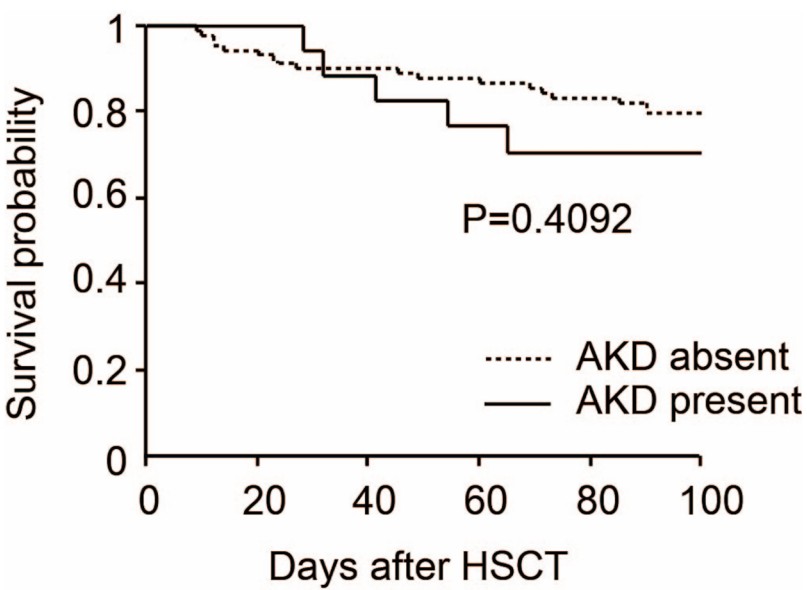

**Figure 1 Overall survival by acute kidney disease after hematopoietic stem cell transplantation.**

incidence of AKD after HSCT was 15.7%. Furthermore, ABO-incompatible HSCT and the incidence of acute GVHD were independently associated with AKD. However, we could not find a significant association between AKD and mortality after HSCT.

Previous reports indicated that the incidence of AKI among HSCT recipients varies widely from 20% to 92% (*Kogon & Hingorani, 2010*; *Clajus et al., 2012*). This difference has varied significantly because of the variations in AKI definitions. Especially, most previous studies defined AKI using simply the doubling of serum creatinine, and this could exclude early stage AKI. KDIGO proposed a new AKI definition which has been integrated from AKIN and RIFLE (*Kidney Disease Improving Global Outcomes (KDIGO), 2012*). However, despite using the KDIGO definition, acute GVHD-related kidney injury could not be evaluated properly, because most acute GVHDs occur within 100 days after HSCT. There were no accurate urinary volume data in our study, but when evaluating AKI according to serum creatinine level alone, only two cases (stages one and three) could be diagnosed as AKI. These results also support that the conventional AKI definition could not be suitable for acute GVHD-related renal injury.

Incidence of AKD in our study was lower than previous reports. Several reasons could be considered. First, medications that can decrease glomerular perfusion, such as renin-angiotensin system inhibitors (RASIs) or NSAIDs were used in our study very few; RASIs were used in two cases (1.9%) and NSAIDs were used in four cases (3.7%), respectively. Second, diabetes which could exacerbate renal function was recognized only nine cases (8.3%) in our study. Third, as previously indicated, early involvement of the nephrologist in our case might prevent incidence of AKD (*Kogon & Hingorani 2010*; *Krishnappa et al., 2016*).

Induction of combination therapy cyclosporine or tacrolimus, calcineurin inhibitors (CNIs), and methotrexate could be superior to CNIs alone in the prophylaxis of

acute GVHD. Although there was no correlation between the incidence of AKD and the use of CNIs in our study, several mechanisms by which cyclosporine use leads to renal injury are proposed. Previous reports indicated that CNIs induced glomerular endothelial dysfunction and renal artery smooth muscle cell contraction, leading to AKD (*Krishnappa et al., 2016*). Our study showed no correlation between CNIs and incidence of AKD. Much of the renal toxicity of CNIs is thought to be dose dependent. One possibility is that CNIs levels in the blood were monitored strictly to prevent side effects in our study. CNIs could activate renin-angiotensin II system (RAS), which increases phospho-c-Raf (Ser338) and Erk1/2 phosphorylation increasing extracellular matrix (*Mima et al., 2012*). RAS also causes vasoconstriction of renal arterioles, inducing AKI (*Menzies et al., 2015*; *Pang et al., 2015*).

The association between AKI and the incidence of GVHD has been reported. After HSCT, kidneys are exposed to GVHD-induced systemic inflammation. Donor T-cells and endothelial dysfunction are also key aspects of developing AKI in GVHD (*Krishnappa et al., 2016*). Further, increases in several cytokines could trigger kidney injury in acute GVHD; when acute GVHD occurred, interleukin (IL)-2, IL-6, or tumor necrosis factor-α was upregulated in the kidney (*Symington et al., 1992*; *Xun et al., 1995*; *Yang et al., 2017*). These cytokines are produced during the initial step of GVHD, resulting in AKD. Furthermore, prolonged increases in these inflammatory cytokines could lead to CKD (*Mima et al., 2012*; *Shlipak & Day 2013*). Another important mechanism of developing GVHD is invasion of inflammatory cells, such as macrophages and monocyte in various organs. Previously, mononuclear cell or T-cell infiltration of renal vessels was reported to be associated with GVHD (*Xun et al., 1995*; *Lo et al., 2002*). GVHD or cytokines released from GVHD could induce endothelial dysfunction, resulting in thrombotic microangiopathy (TMA). TMA includes thrombocytopenic purpura and hemolytic uremic syndrome. Generally, HSCT-induced TMA is not related to deficiencies in von Willebrand factor-cleaving protease, but to endothelial dysfunction from GVHD-induced direct cell injury or by several cytokines derived from GVHD occurring in other parts of the body.

The relationship between ABO-incompatible HSCT and AKD is still unclear. However, ABO incompatibility could induce GVHD; ABO antigens are recognized in several organs including endothelial cells where donor origin blood cells might attack, triggering GVHD (*Seebach et al., 2005*; *Stussi et al., 2002*). Thus, there is a possibility that ABO incompatibility-induced GVHD is a major risk of AKD in our study rather than ABO incompatibility and GVHD respectively.

Hepatic sinusoidal obstruction syndrome (SOS), previously known as veno-occlusive disease, is one of the major complications after HSCT and could be an independent risk factor for AKD development. The mechanisms of SOS-induced AKD are still unclear, but one possibility is that SOS may exacerbate liver function itself, resulting in hepatorenal syndrome. Moreover, some reports have suggested that endothelial dysfunction is essential for SOS development. We have shown that plasminogen activator inhibitor (PAI)-1 was increased in endothelial dysfunction (*Mima et al., 2012*).

Furthermore, previous reports indicated that the levels of PAI-1 expression could be diagnostic and predicting marker for SOS (*Salat et al., 1997*; *Pihusch et al., 2005*). In addition, use of TBI > 12 Gy could be one of the risk factors for SOS (*Richardson et al., 2012*). In the present study, nine patients underwent total body irradiation with more than 12 Gy, but development of SOS was not recognized in any patients.

Despite the absence of data on endothelial function, such as PAI-1, von Willebrand factor, or a disintegrin-like metalloproteinase with thrombospondin type 1 motifs 13 (ADAMTS 13) in our study, endothelial dysfunction might be induced by HSCTs among the AKD patients. Previous reports indicated that ABO-incompatible HSCT increased the incidence of SOS, leading to AKD (*Benjamin & Antin, 1999*; *Benjamin et al., 1999*). Our data indicated that ABO-incompatible or GVHD-related AKD did not affect the overall survival. Similar to our results, several previous reports showed an increased mortality in ABO-incompatible transplants without an effect on overall survival (*Benjamin & Antin, 1999*; *Benjamin et al., 1999*), while *Kimura et al. (2008)* showed lower overall survival and increased mortality rates in ABO-mismatched transplants. Thus, it is still unclear whether ABO-incompatible HSCT increases AKD and mortality rates.

There might be some limitations to our study, because this was a retrospective analysis and a single-center study. Several reasons could be considered. First, we could not adequately examine urine volume associated with diagnosis of AKI and we used serum creatinine as the principal criterion in the definition of AKD. Second, we did not have data on cytomegalovirus infection that often recognized in HSCT and can increase inflammation in kidney. Third, we did not have precise data on severity of acute GVHD. Lastly, we did not have renal pathological data that could help to enhance our understanding of the etiology of AKD. We would like to improve the study protocol to include periodic urinalysis and renal biopsies.

## CONCLUSIONS

There is an association between ABO-incompatible HSCT and AKD as well as an association between HSCT related acute GVHD and AKD. GVHD and ABO-incompatible HSCT emerges as an important cause of AKD regardless of nephrotoxic agents. Our data also demonstrated that mortality was not increased with the incidence of AKD.

## ACKNOWLEDGEMENTS

Part of this manuscript was presented at the 2nd Asia Pacific AKI CRRT Congress 2018. We wish to thank Hitoshi Hanamoto, Hideo Yagi, and Mariko Fujita for consultation on the work-up of the patients and this study.

### Funding

This work was supported by JSPS KAKENHI Grant Number 17K09720. The funders had no role in study design, data collection and analysis, decision to publish, or preparation of the manuscript.

## Grant Disclosures

The following grant information was disclosed by the authors:
JSPS KAKENHI: 17K09720.

## Competing Interests

The authors declare that they have no competing interests.

## Author Contributions

- Akira Mima conceived and designed the experiments, performed the experiments, analyzed the data, contributed reagents/materials/analysis tools, prepared figures and/or tables, authored or reviewed drafts of the paper, approved the final draft.
- Kousuke Tansho performed the experiments, analyzed the data, contributed reagents/materials/analysis tools, prepared figures and/or tables.
- Dai Nagahara performed the experiments, contributed reagents/materials/analysis tools, prepared figures and/or tables.
- Kazuo Tsubaki conceived and designed the experiments, authored or reviewed drafts of the paper.

## Human Ethics

The following information was supplied relating to ethical approvals (i.e., approving body and any reference numbers):

The clinical study ethics review board of Kindai University Nara Hospital (approval number: 18-6) approved this study.

## Data Availability

The raw measurements are available in a Supplementary File.

## Supplemental Information

Supplemental information for this article can be found online at http://dx.doi.org/10.7717/peerj.6467#supplemental-information.

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
