# Peer review of "Incidence of acute kidney disease after receiving hematopoietic stem cell transplantation: a single-center retrospective study"

_PeerJ, doi:10.7717/peerj.6467_

## Round 0.1 · original submission · Major Revisions

Please fully address all comments from both reviewers below.

·

Basic reporting

I would like to commend the authors on an interesting study on the risk factors for acute kidney disease in patients receiving hematopoietic stem cell transplantation. The study is generally clear, unambiguous, and well written. The background could include more introduction to the different known risk factors for acute kidney disease among patients receiving hematopoietic stem cell transplantation. The structure of the article conforms to an acceptable format. Figure 1 is readable, although the lines could be thicker. Also, there is a small fragment of the Kaplan Meier curve below the number “20” on the x-axis.
The raw data is individual based micro data. I am not aware of the legal basis of sharing data in Japan, but where I come from it would be illegal to share such detailed information.

Experimental design

The background introduces the reader to the concept of acute kidney disease, and the possible advantage of using this, compared to acute kidney injury, to assess the incidence of decreased renal function after receiving hematopoietic stem cell transplantation. One important thing missing, namely a clear aim of the study. The study aim is reported by the authors “…to determine the incidence of AKD after HSCT”. However, the title claims otherwise. Please clarify and amend accordingly. The authors do not clearly states how the research fills an identified knowledge gap.
The methods section is clear, concise, and described in detail. It would be possible to reproduce the study. The authors have chosen to perform univariate statistics, which is applauded given the low number of outcomes. However, some questions regarding the statistical analyses should be answered:
1 – The authors state that continuous variables were presented as medians. How was the variance presented? As interquartile range? In the results section (line 105) no variance in age is presented. And more importantly in table 1 the age is presented as 46 +/- 15 years. The discrepancy is most likely that the authors present age in mean and standard deviation there. Please amend to median with IQR. Also, the authors state (line 92) “…compared using the Pearson’s chi-squared test. Fisher’s exact tests were used.” This is not intuitive. One could guess, that the authors used Fischer’s exact test, when the numbers were below X. Please clarify.

Validity of the findings

The authors use much of the discussion (from line 144-151 and 164-174) to discuss topics not directly relatable to their results. Two potential risk factors were identified (ABO incompatible and acute GVHD), all other listed variables were not associated with acute kidney disease. The authors should discuss their findings and relate them to the literature.
The limitations section needs to be amended. The authors could discuss the low number of outcomes and the appertaining lack of multivariable analysis. Also, the impact of the two listed limitations (retrospective analysis and single-center study) should be discussed more in detail.
I do not agree with the conclusion. The results do not offer the probabilities of finding out which ABO-incompatible HSCT or HSCT related acute GVHD patients are more prone to developing AKD, but rather that there is an association with ABO-incompatible HSCT and AKD as well as an association with HSCT related acute GVHD and AKD.

Reviewer 3 ·

Basic reporting

This is a generally well-written single center retrospective study of predictors of acute kidney disease post hematopoetic stem cell transplantation. The authors propose a new concept, acute kidney disease (AKD), and seem to suggest this may be a better way to characterize kidney injury in HSCT patients.

Some issues to consider-
1. The KDIGO AKI definition is never fully stated. Instead, they reference that use of KDIGO guidelines based on creatinine only. On my brief review, KDIGO guidelines include:
1. increase in SCr >/= 0.3 within 48 hours or
2. increase in SCr >/== 1.5x baseline within 7 days or
3. urine volume <0.5ml for 6 hours.

Which creatinine based criteria did they follow?

2. More rationale for the "new" definition of AKD is required. There is mention that this will allow better characterization of kidney injury during the first 3 months when GVHD may also occur but the authors should elaborate more. Also, there are renal manifestations of acute GVHD itself which should be mentioned (Eg. nephritis or nephrotic syndrome). Are there any references to support the new AKD definition?

3. Minor grammatical corrections:
- last sentence of methods section capitalized
- first sentence of introduction is vague. Consider instead "HSCT is a fundamental therapy for high risk and refractory hematological diseases. Several types of HSCT procedures exist based on conditioning therapy (myeloablative or reduced intensity), donor source (autologous and allogeneic, including matched related, matched unrelated, haploidentical and umbilical cord) and stem cell source (peripheral blood stem cells or bone marrow stem cells). Some of this is highlighted in table 1. which needs a key to abbreviations used below it.

4. Just out of curiosity, did any patients require renal replacement therapy (dialysis or CVVH?) This should be reported.

5. In the discussion, it should be explained why the rate of AKD (15.7%) is lower than that reported in other studies using AKI as definition (20-92%).

Experimental design

Standard retrospective review with appropriate statistical analysis.

Validity of the findings

No comment

Additional comments

See basic reporting section.

---

## Round 0.2 · Minor Revisions

Dear Dr. Mima,

My name is Stefano Menini. I serve as Academic Editor for PeerJ in the subject area of Nephrology. I was asked to step in to make this decision as the previous Editor is unavailable.

I carefully read your manuscript entitled "Incidence of acute kidney disease after receiving hematopoietic stem cell transplantation: a single-center retrospective study" and all the reviewers' comments. Basically the revision is now acceptable for publication, but before final acceptance is given, I would appreciate it if you would address the remaining issue on minor persistent grammar mistakes raised by Reviewer 2.

If you are willing to do this, it would not be necessary for me to return the manuscript to the reviewers, but it could then be accepted for publication.

Sincerely yours,

Stefano Menini

Reviewer 3 ·

Basic reporting

The authors have addressed my concerns and only minor grammatical errors persist. Please run an English spell and grammar check to fix these.

Experimental design

No issues.

Validity of the findings

No issues.

Additional comments

The authors have addressed my concerns and only minor grammatical errors persist. Please run an English spell and grammar check to fix these.

---

## Round 0.3 · accepted · Accept

Dear Dr. Mima,

I am pleased to inform you that the revision of your manuscript entitled "Incidence of acute kidney disease after receiving hematopoietic stem cell transplantation: a single-center retrospective study" now makes it acceptable for publication in PeerJ. I appreciate very much your making the suggested revisions. The manuscript will now be forwarded to the product editor for copy editing and publication.

Sincerely yours,

Stefano Menini